

# In vitro and in vivo hypolipidemic properties of the aqueous extract of *Spirulina platensis*, cultivated in colored flasks under artificial illumination

Mahmoud A. Al-Saman[1], Nada M. Doleib[2,3], Mohamed R. Ibrahim[1], Mohamed Y. Nasr[4], Ahmed A. Tayel[5] and Ragaa A. Hamouda[2,6]

[1] Department of Industrial Biotechnology, Genetic Engineering and Biotechnology Research Institute, University of Sadat City, Sadat City, Egypt
[2] Department of Biology, Faculty of Sciences and Arts Khulais, University of Jeddah, Jeddah, Saudi Arabia
[3] Department of Microbiology, Faculty of Applied and Industrial Science, University of Bahri, Khartoum, Sudan
[4] Department of Molecular Biology, Genetic Engineering and Biotechnology Research Institute, University of Sadat City, Sadat City, Egypt
[5] Faculty of Aquatic and Fisheries Sciences, Kafrelsheikh University, Kafrelsheikh, Egypt
[6] Department of Microbial Biotechnology, Genetic Engineering and Biotechnology Research Institute, University of Sadat City, Sadat City, Egypt

Corresponding author
Ragaa A. Hamouda,
ragaa.hamouda@gebri.usc.edu.eg

## ABSTRACT

**Background:** *Spirulina* is blue-green algae that grows mainly in tropical and subtropical lakes and is commonly used due to its nutritional features including high concentrations of protein, vitamins, mineral salts, carotenoids and antioxidants. This study aimed to investigate the anti-hypercholesterolemic potential of aqueous extract of *Spirulina platensis* cultivated in different colored flasks under artificial illumination; in vitro and in the diet induced hypercholesterolemic Swiss albino mice.

**Methods:** *Spirulina platensis* was cultivated in red, blue, green and colorless Erlenmeyer flasks containing Zarrouk's medium under aerobic conditions, with incessant illumination by artificial cool white fluorescent with light intensity of 2500 lux (35 µmol photon m$^{-2}$ s$^{-1}$). Chlorophyll a and total carotenoid contents were estimated using colorimetric methods, fatty acids composition was determined by GC-Mass, in vitro and in vivo anti-cholesterol assays were used in assessing the anti-hypercholesterolemia potential of obtained *Spirulina* cells.

**Results:** The results showed that the highest cell dry weight, chlorophyl a, and carotenoid of *S. platensis* were observed in colorless flasks and that the lowest values were recorded with the green colored flasks. Also, the hot water extract of *S. platensis* obtained from colorless flasks at a concentration of 15 mg/mL after 60 min of incubation exhibited the greatest reduction of cholesterol level. Gas chromatography-mass spectrometry analysis of *S. platensis* methanolic extract showed 15 bioactive compounds were identified and grouped according to their chemical structures. An experimental model of hypercholesterolemic mice had been examined for impact of *S. platensis* individually and combined with atorvastatin drug. All *S. platensis* groups resulted in a remarkable decrease in plasma total cholesterol, triglycerides and low density lipoprotein; and increase in high density lipoprotein.

**Conclusion:** The present study concluded that the hot aqueous extract of *S. platensis* developed in colorless flasks is recommended as a natural source for bioactive compounds, with anti-cholesterol and antioxidant potentialities.

## INTRODUCTION

Cholesterol is an important organic molecule of the cell membranes, and a precursor for the biosynthesis of bile acids, steroid hormones, and vitamin D. Cholesterol plays an essential role in human heart health. High-density lipoprotein (HDL) is considered as "good cholesterol" and low-density lipoprotein (LDL) is known as "bad cholesterol". High cholesterol in serum is a preceding risk factor for human cardiovascular disease (CVD) (*Tabas, 2002*). CVD for instance coronary heart disease is one of the major causes of death and other disabilities in developing countries (*Nascimento et al., 2002*).

Cholesterol in the human body is formed in the liver or straightly absorbed from the diet such as animal fat-based foods but in different quantities (*Christie, 2003*). Most ingested cholesterol is esterified within 7–10 h after ingestion (*Soliman, 2018*), which causes it to be poorly absorbed by the gut. The body also compensates for absorption ingested cholesterol by reducing its own cholesterol synthesis (*Lecerf & De Lorgeril, 2011*).

Various drugs are used to lower blood cholesterol levels. Statins are a widely prescribed class of drugs to lower cholesterol, include for example atorvastatin, lovastatin and simvastatin. Statins inhibit the de novo synthesis of cholesterol via inhibition of HMG-CoA reductase enzyme in the liver that is responsible for making cholesterol (*Ward, Watts & Eckel, 2019*). However, such drugs are expensive, and undesirable side effects have been reported, such as muscle pain, liver inflammation, type II diabetes and neurological side effects. This has logically allowed patients to search for alternative safe options, such as weight reduction, use of dietary supplements, fat-free diets and exercise (*Michael et al., 2016*).

Algae are photosynthetic organisms (photoautotrophs) that are capable of photosynthesis by sunlight or artificial light. Microalgae have higher growth rates, and larger quantities of high-value products, such as pigments and dietary supplements (*Zeng et al., 2011*). Therefore, at present microalgae are under limelight worldwide due to its importance. The cultivation of algae can be done in open systems like raceway ponds (*Vardaka et al., 2016*), or in closed photobioreactors for instance, vertical-column, flat plate and tubular photobioreactors (*Sánchez Mirón et al., 2002*; *Samson & Leduy, 1985*; *Kaewpintong et al., 2007*); and both are used commercially. Most algal culture systems rely on Beer–Lambert's law to determine the light intensity ($I$) depending on the depth of culture and the concentration of biomass. High values of $I$ promote growth parameters,

while low values result in a biomass that is rich in pigments and proteins (*Soni, Sudhakar & Rana, 2017*).

Artificial light provides better regulation of intensity, duration, and light spectra, usually involves fluorescent lamps, incandescent, or halogen lights which have wide emission spectra. Light characteristics have a profound influence on microalgae metabolism and development. *Barufi, Figueroa & Plastino (2015)* reported that growth rates were higher in *Gracilaria birdiae* exposed to white light, and *Sharmila et al. (2018)* concluded that the yellow color light could be used for the more lipid production, whereas blue light for biomass and pigments.

Spirulina is a blue-green algae that grows primarily in tropical and subtropical lakes and is widely used due to its nutritional features including high concentrations of protein (~60%) of its dry weight, vitamins ($B_1$, $B_2$, $B_3$, $B_6$, $B_9$, $B_{12}$, C, D and E), mineral salts (mainly, potassium and iron), carotenoids, and antioxidants (*Gutiérrez-Salmeán, Fabila-Castillo & Chamorro-Cevallos, 2015*). The United Nations world at food conference declared that Spirulina as the best food for future (*Pulz & Gross, 2004*). *Spirulina platensis* had the pharmacological analyses which revealed the beneficial properties in both in vitro and in vivo, including antioxidant (*González de Rivera et al., 1993*), immunomodulation (*Yang, Lee & Kim, 1997*; *Kim et al., 1998*) antiviral (*Abed, Dobretsov & Sudesh, 2009*), anticancer (*Gantar & Svirčev, 2008*), cholesterol reduction (*Ponce-Canchihuamán et al., 2010*), and anti-diabetes (*Layam & Reddy, 2006*) activities. *S. platensis* is the heaviest algal source of Gamma-linolenic acid (γ-linolenic acid; GLA), a precursor for the biologically-active compounds (prostaglandins, PG) (*Habib et al., 2008*).

Based on, the previous studies that assume that algae may play a critical role in regulating lipid metabolism in animals and humans and other results that suggest that color lights have influence on regulating algal growth, and photosynthetic pigments synthesis. The purpose of this experimental study is to grow *S. platensis* in flasks of different colors (Red, blue, green and colorless) using Zarrouk's medium at constant light intensity

   i) To analyze the different growth parameters such as biomass productivity and pigment content under artificial illumination.
   ii) To study the influence of the hot water extract of *S. platensis* biomass, obtained from the colored flasks on cholesterol levels reduction in vitro and in vivo animal model.
   iii) To extract and analysis of fatty acids composition of *S. platensis* by gas chromatography-mass spectrometer (GC-Mass).

## MATERIALS AND METHODS

### Algal cultures

Pure culture of *Spirulina platensis* was obtained from the Microbiology Department, Genetic Engineering and Biotechnology Research Institute, University of Sadat City, Egypt (*Hamouda, Sorour & Yehheia, 2016*).

**Table 1 Modified constituents of Zarrouk's medium.**

| Constituent A | g/L |
| --- | --- |
| $NaHCO_3$ | 18 |
| $NaNO_3$ | 2.5 |
| $K_2SO_4$ | 1 |
| NaCl | 1 |
| $K_2HPO_4$ | 0.5 |
| $MgSO_4 \cdot 7H_2O$ | 0.2 |
| $Na_2EDTA$ | 0.08 |
| $CaCl_2$ | 0.04 |
| $FeSO_4 \cdot 7H_2O$ | 0.01 |
| Constituent B | 1 ml/L |
| Trace elements | g/L |
| $H_3BO_3$ | 2.860 |
| $MnCl_2 \cdot 4H_2O$ | 1.800 |
| $ZnSO_4 \cdot 7H_2O$ | 0.220 |
| $Cu_2SO_4$ | 0.080 |
| $(NH_4)_6Mo_7O_{24} \cdot 4H_2O$ | 0.020 |

## Culture maintenance

The blue green alga *S. platensis* was sustained on Zarrouk's medium (*Zarrouk, 1966*; *Belay, 1997*) having the subsequent constituents Table 1:

## The effect of colored flasks on cell dry weight

Red, blue, green and colorless Erlenmeyer flasks containing Zarrouk's medium were used for culturing *S. platensis*. All flasks including 100 mL of selected media under aerobic conditions, were autoclaved for 20 min at 121 °C, then incubated at 26 ± 2 °C with incessant illumination by cool white fluorescent with light intensity of 2,500 lux (35 μmol m$^{-2}$ s$^{-1}$), after that left to grow with two times daily shaking to avoid algal cell clumping and adherence of algal cells to the containers. Cultures were gathered after 2 weeks by centrifugation at 5,000 rpm (Centurion Scientific LTD Model 1020 series) for 10 min and washed well with distilled water (*Hamouda, Sorour & Yehheia, 2016*).

The algal biomass was evaluated by measuring its dry weight. All samples were filtered using filter paper (pore size 8 mm), washed with distilled water, and dried at 60–65 °C till constant weights (*Nigam et al., 2007*).

## Pigments estimation

A known volume of *S. platensis* culture was centrifuged at 8,000 rpm for 10 min, and then *S. platensis* pellets were treated with the same volume of 90% acetone, kept in water bath for 30 min at 55 °C, and followed by centrifugation once again at 8,000 rpm for 10 min. The color of pellets must be white to guarantee complete extraction of pigments, after that supernatant was collected and then the absorbance was measured at three different wavelengths 663, 645 and 480 nm. Calculations were made according to the

formula devised by the method of *Jeffrey & Humphrey (1975)* for chlorophyl a, and *Ridley (1977)* for total carotenoid estimation.

Chlorophyll a = [12.7 $A_{663}$ − 2.69 $A_{645}$] × vol. of extraction/weight of the sample
where $A_{663}$ is the absorbance at 663 nm and $A_{645}$ is the absorbance at 645 nm.

Total carotenoid (mg/g) = 4 × $A_{480}$ × vol. of extraction/weight of the sample
where $A_{480}$ is the optical density at 480 nm and 4 is the correction factor.

## Preparation of algal extracts and in vitro anti-cholesterol assay

Algal extracts were prepared according to *Hamouda et al. (2017)*. Different amounts of dried algal biomass (50, 100, 150 and 200 mg) were boiled in 10 mL distilled water for 30 min, cooled and centrifuged at 1,000 rpm for 5 min, after that the supernatants were taken as the algae extracts.

One hundred μL of each algae extracts was added to 100 μL of cholesterol standard (The cholesterol 200 mg was dissolved in 100 mL of phosphate buffer, pH 7.0 and 1% of Triton X-100 surfactant), mixed and incubated for different periods (15, 30, 45, 60 min) at 37 °C. The cholesterol concentration was measured using enzymatic colorimetric method according to *Richmond (1973)*, using cholesterol kit "Cholesterol–Liquizyme (Elitech, Puteaux, France)". The contents were incubated for 10 min at room temperature and the absorbance of sample and standard were read at wavelength 500 nm in a UV-Vis spectrophotometer against reagent blank. Anti-cholesterol activity of the extract was calculated using the following equation:

$$\text{Inhibition}(\%) = \text{Standard conc.} - \text{Sample conc.}/\text{Standard conc.} \times 100$$

## Extraction and determination of fatty acid composition

One g of dry powder of *S. platensis* was extracted twice with 20 mL methanol 99.5%, and was stirred with magnetic stirrer for 15 min; then the supernatant was collected by filtration. All supernatants were collected as a bold fraction and the solvent was evaporated using rotary evaporator. The methyl esters can be attained by transmethylation of the lipids by refluxing them for 90 min with methanol-benzene-sulfuric acid (20:10:1), respectively (*Harborne, 1973*; *Mendham et al., 2000*). The solution was concentrated to two thirds of its volume, water was then added for washing till free from acidity as indicating by litmus paper indicator, dried over anhydrous sodium sulfate and filtered, and the chemical composition was performed using Trace GC-ISQ Q mass spectrometer (Thermo Scientific, Austin, TX, USA) according to the method described by *El-Dougdoug et al. (2018)*.

## Feeding study

### The animal model and experimental design

Male Swiss Albino (SA) mice, 2 weeks old were purchased from VACSERA (Egyptian Company for Production of Vaccines, Giza, Egypt). The animals were housed in polycarbonate cages in a conditioned room at 20 °C ± 2 with illumination for 12 h.

**Table 2 Experimental groups of mice and diets used in the feeding study.**

| Groups | Diet | Formulation |
|--------|------|-------------|
| I | Cholesterol-free diet | 1 Kg basal diet* + 500 mL water |
| II | *S. platensis* | 1 Kg basal diet + 500 mL water + *S. platensis* (15 mg/mL) |
| III | Cholesterol-enriched diet | 1 Kg basal diet + 500 mL water + 1 g cholesterol |
| IV | Cholesterol-enriched diet + *S. platensis* | 1 Kg basal diet + 500 mL water + 1 g cholesterol + *S. platensis* (15 mg/mL) |
| V | Cholesterol-enriched diet + atorvastatin | 1 Kg basal diet + 500 mL water + 1 g cholesterol + 10 mg/Kg atorvastatin |
| VI | Cholesterol-enriched diet + *S. platensis* + atorvastatin | 1 Kg basal diet + 500 mL water + 1 g cholesterol + *S. platensis* (7.5 mg/mL) + 5 mg/Kg atorvastatin |

Note:
* Basel diet composition according to *Reeves, Roossow & Lindlauf (1993)*.

The initial body weight of the animals was approximately 30 g. Weight gain was monitored and the 24 h food intake was recorded every 5 days. The duration of experimental treatment was 5 weeks. All animals were allowed to adapt to the environment for at least 10 days prior to dietary treatment. All the experimental procedures were carried out in accordance with international guidelines for care and use of laboratory animals and Ethics Committee of the Genetic Engineering & Research Institute, Sadat City University, Egypt. (Approval number: gebri USC-009-1-19).

The animals were randomly divided into six groups and each group containing 8 mice; two control groups and four treatment groups respectively according to *Lar, Patel & Pandanaboina (2016)* with minor modifications.

**Group I:** negative control (was fed on the basic diet according to *Reeves, Roossow & Lindlauf (1993)* plus 1% saline (oral gavage)). **Group II:** was fed regular diet with *S. platensis* water extract in concentration of 15 mg/mL (1% of body weight) administration by oral gavage once daily for 5-week period. **Group III:** positive control (was fed a high fat diet for 5-week period). **Group IV:** was fed a high-fat diet with *S. platensis* water extract in concentration of 15 mg/mL for 5-week period. **Group V:** was fed a high-fat diet with 10 mg standard atorvastatin for 5-week period. **Group VI:** was fed a high-fat diet with *S. platensis* water extract in concentration of 7.5 mg/mL plus 5 mg standard atorvastatin.

The composition of diets and animal groups is presented in Table 2. Body weights were measured at the end of the experimental period.

### Blood sampling from sacrificed mouse and serum analysis

All the animals were euthanized using diethyl ether as an inhalant anesthesia and blood samples were drawn from the heart of each sacrificed mouse. The blood samples were collected in test tubes for determination of biochemical analyses (serum lipid profile concentrations, liver enzymes activities, serum creatinine, and urea). The liver and heart were carefully dissected out, weighed and immediately preserved in 10% neutral buffered formalin. Blood serum was separated by centrifugation at 3,000 rpm for 15 min to obtain plasma, which was kept frozen at −20 °C until analysis.

i) **Lipid profile analysis**

Total cholesterol (TC) in serum, HDL cholesterol and triglycerides (TG) were determined according to *Lopes-Virella et al. (1977)* and *Fossati & Prencipe (1982)*, respectively. LDL cholesterol was calculated according to *Warnick et al. (1990)* as follows: LDL = TC − HDL − (TG/5).

ii) **Kidney functions**

The urea and creatinine concentrations were measured using colorimetric enzymatic method according to *Kaplan & Glucose (1984)* and *Murray (1984)* respectively, by use of Diamond Diagnostics—Egypt kit.

iii) **Liver functions**

Aspartate aminotransferase (AST/SGOT) and alanine aminotransferase (ALT/SGPT) were measured using colorimetric enzymatic methods according to *Reitman & Frankel (1957)* by use of Spinreact—Spain kit.

## Histopathological examination

Liver and heart specimens were sliced, and pieces were preserved in 10% formalin for proper fixation. These tissues were processed and embedded in paraffin wax. Sections of 4–5 microns in thickness were cut and stained with hematoxylin and eosin. All the stained sections of the tissues were examined under microscope for circulatory disturbances, inflammation, degenerations, apoptosis, necrosis, and any other pathological changes, according to the method of (*Suvarna, Layton & Bancroft, 2013*).

## Statistical analyses

All data were subjected to analysis of variance (ANOVA). Three replicates of each item were analyzed and the main values as well as the SD were given. Significance of the variable mean differences was determined using Duncan's multiple range tests ($p \leq 0.05$).
All analyses were carried out using SPSS 16 software (*Sokal & Rohlf, 1995*).

## RESULTS

### Estimation of algal growth, chlorophyl a and carotenoid

Results in Table 3 A represent the effect of cultivation of blue-green algae *S. platensis* in flasks with different colors (colorless, red, green and blue) on its cell dry weight, chlorophyl a and total carotenoid contents.

The highest cell dry weight of *S. platensis* (3.40 g/L) was observed using the colorless flasks that was followed by (2.60 and 2.18) g/L using the red and blue colored flasks, respectively. On the other hand, the green-colored flasks gave the lowest cell dry weight with an average of 2.00 g/L.

*Spirulina platensis* cultivated in colorless flasks demonstrated the highest chlorophyl a content 17.85 mg/g (1.78%) based on dry weight, while *S. platensis* cultivated in green-colored flasks had the lowest chlorophyl a content 10.50 mg/g (1.05%). On the other hand, the highest contents of carotenoid 61.88 mg/g (6.19%) was detected in dried cells of cultivated *S. platensis* in colorless flasks, followed by 47.32 mg/g (4.73%) using the red-colored flasks and 39.68 mg/g (3.97%) using the blue-colored flasks, respectively.

**Table 3** The effect of cultivation in colored glass bottles on the growth performance, total carotenoid and chlorophyl a content of S. platensis.

| Flask color | Cell dry weight (g/L) | Chlorophyll a | | Total carotenoid | |
|---|---|---|---|---|---|
| | | mg/g (dry weight) | % (dry weight) | mg/g (dry weight) | % (dry weight) |
| Colorless | 3.40 | 17.85 | 1.78 | 61.88 | 6.19 |
| Red | 2.60 | 13.65 | 1.36 | 47.32 | 4.73 |
| Blue | 2.18 | 11.45 | 1.14 | 39.68 | 3.97 |
| Green | 2.00 | 10.50 | 1.05 | 36.40 | 3.64 |

While the green-colored flasks, gave the lowest carotenoid content with an average of 36.40 mg/g (3.64%). There are variations in pigment content in all experimental trials, which may be associated with color of flasks.

## In vitro cholesterol-lowering activity of *Spirulina platensis* cultivated in colored flasks

To investigate the potential cholesterol-lowering effect of S. platensis; we carried out a screening on the cholesterol-lowering abilities of different hot water extracts of dried S. platensis obtained from different colored glass flasks. An anti-cholesterol assay was performed and the results revealed the potential activity of all extracts of algae in a dose-dependent manner.

Figure 1A shows the percentages of the cholesterol reducing activity of water extract of S. platensis cultivated in colorless flasks as a control treatment. The highest cholesterol reduction percentage (76.01%) was observed using the hot water extract of S. platensis at a concentration of 15 mg/mL after 60 min of incubation that was followed by 5 mg/mL (74.64%) followed by 20 mg/mL (72.27%) after incubation for time intervals of 60 and 30 min, respectively. On the other hand, the concentration of 5 mg/mL gave the lowest cholesterol reduction percentage with an average of 61.85% value ($p > 0.05$) after a period of 15 min. No significant differences were observed between the treatments, furthermore the S. platensis concentration and incubation time had no significant effect on cholesterol-lowering levels. Figure 1B illustrates the percentages of the cholesterol reduction of S. platensis obtained from blue flasks. No significant differences were observed between the treatments, moreover the S. platensis concentration and incubation time had no significant effect on cholesterol-lowering levels; and the reduction rate ranged between 45.5% and 48.4%. Figure 1C illustrates the percentages of the cholesterol reduction at different periods of time and different concentrations of dry weight of S. platensis obtained from green flasks. The highest cholesterol reduction percentage (52.8%) was observed using the hot-water extract of S. platensis at a concentration of 15 mg/mL after 60 min of incubation; on the other hand, the concentration of 20 mg/mL gave the lowest cholesterol reduction percentage with an average of 40.9% value ($p < 0.05$) after a period of 15 min. No significant differences were observed between the treatments, furthermore the S. platensis concentration and incubation time had no

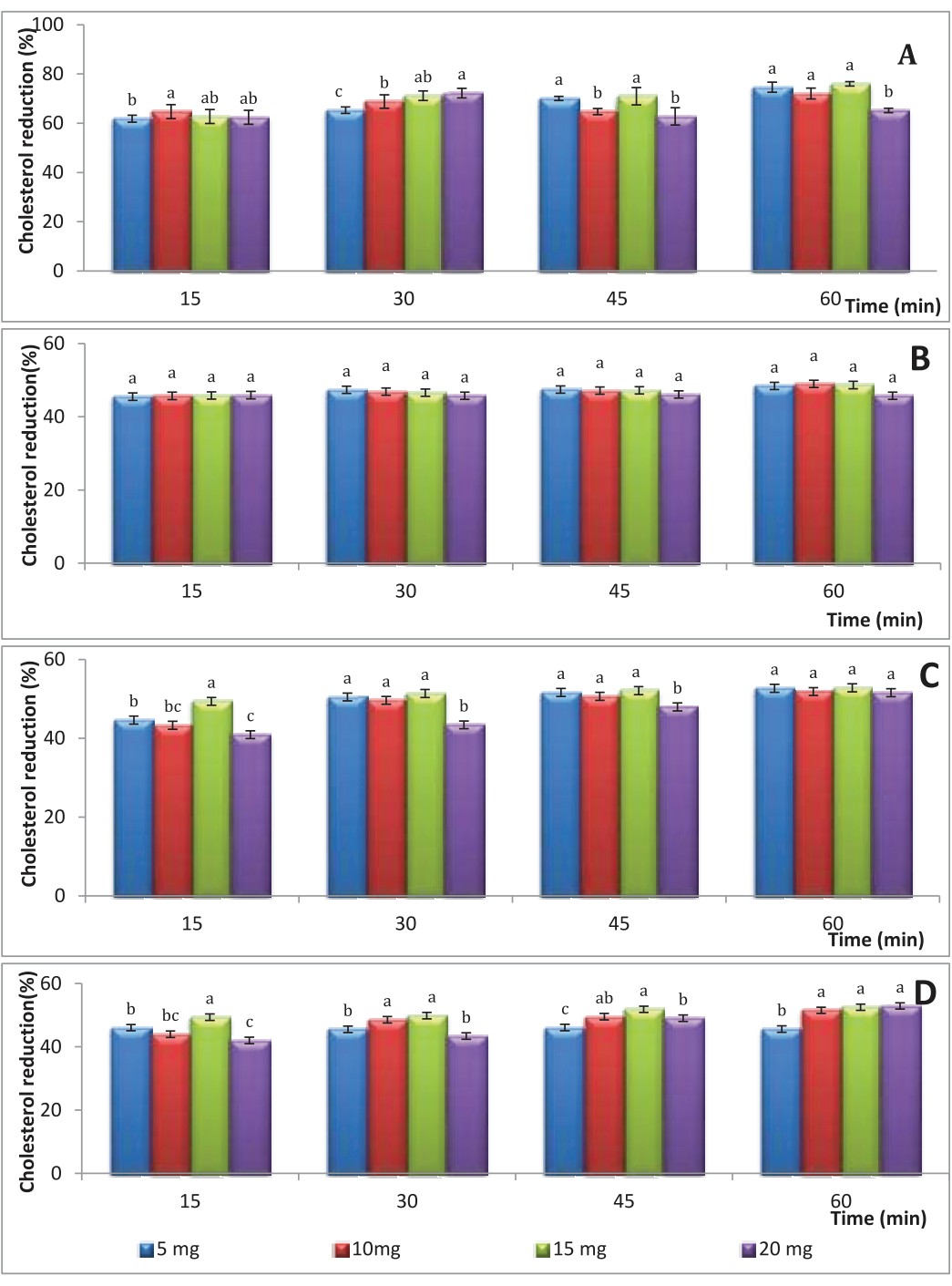

**Figure 1 The effect aqueous extracts of *S. platensis* (mg/mL) obtained from colored flasks on total cholesterol; (A) control; (B) blue; (C) green; (D) red; the different super script letters in the same column group are significantly different (*p* < 0.05).**

significant effect on cholesterol-lowering levels. In the case of red flasks (Fig. 1D), the highest cholesterol reduction percentage (52.9%) was observed using the hot-water extract of *S. platensis* at a concentration of 20 mg/mL after 60 min of incubation.
**Table 4 Volatile compounds identified in methanolic extract of *S. platensis* obtained from colorless flasks by GC-MS.**

| Compound name | Molecular formula | RT * (min) | Area (%) | M. Wt |
|---|---|---|---|---|
| Hexamethylcyclotrisiloxane | $[(CH_3)_2SiO]_3$ | 03.79 | 1.23 | 222 |
| propanedioic acid (malonic acid), dimethyl ester | $C_5H_8O_4$ | 06.58 | 0.31 | 132 |
| palmitic acid (hexadecanoic acid), ethyl ester | $C_{18}H_{36}O_2$ | 26.42 | 0.51 | 284 |
| 9-Octadecenoic acid, (2-phenyl-1,3-dioxolan-4-yl) methyl ester, cis- | $C_{28}H_{44}O_4$ | 27.40 | 0.46 | 444 |
| 10,13-octadecadiynoic acid, methyl ester | $C_{19}H_{30}O_2$ | 27.81 | 0.56 | 290 |
| 7,10-pentadecadiynoic acid | $C_{15}H_{22}O_2$ | 28.39 | 0.44 | 234 |
| hi-oleic safflower oil | $C_{21}H_{22}O_{11}$ | 29.41 | 0.66 | 450 |
| 9-octadecenoic acid (Z)-, phenylmethyl ester | $C_{25}H_{40}O_2$ | 29.81 | 0.69 | 372 |
| palmitic acid, (2-phenyl-1,3-dioxolan-4-yl) methyl ester | $C_{26}H_{42}O_4$ | 31.74 | 0.35 | 418 |
| isochiapin b | $C_{19}H_{22}O_6$ | 32.78 | 3.81 | 346 |
| dimethoxyglyceroldocosyl ether | $C_{27}H_{56}O_5$ | 32.92 | 02.1 | 460 |
| oxiraneundecanoic acid, 3-pentyl-, methyl ester | $C_{19}H_{36}O_3$ | 33.39 | 2.32 | 312 |
| oleic acid, eicosyl ester | $C_{38}H_{74}O_2$ | 33.86 | 0.52 | 562 |
| 17-pentatriacontene | $C_{35}H_{70}$ | 34.34 | 0.25 | 490 |
| 9-hexadecenoic acid, eicosyl ester, (Z)- | $C_{36}H_{70}O_2$ | 34.86 | 1.69 | 534 |

**Note:**
   \* Retention time.

## GC-MS analysis of *Spirulina platensis*

Methanolic extract of *S. platensis* cultivated in colorless flasks was analyzed by GC-MS for determining of its volatile components based on their retention time and peak area (Table 4; Fig. 2). The GC-MS analysis showed 15 bioactive compounds were identified and grouped according to their chemical structures. The major volatile components contained were isochiapin B (3.81%), oxiraneundecanoic acid, 3-pentyl (2.32%), dimethoxy glycerol docosyl ether (2.1%), 9 hexadecenoic acid, eicosyl ester (1.69%), hexamethylcyclotrisiloxane (1.23%), 9-octadecenoic acid (Z)-, phenylmethyl ester (0.69), oleic safflower oil (0.66%), palmitic acid (0.51%), 10,13-octadecadiynoic acid (0.56%), 9-octadecenoic acid, (2-phenyl-1,3-dioxolan-4-yl) methyl ester, cis- (0.46%), 7,10-pentadecadiynoic acid (0.44%), propanedioic acid dimethyl ester (0.31%), palmitic acid (2-phenyl-1,3-dioxolan-4-yl) (0.35%).

The presence of antioxidant compounds such as mono and polyunsaturated fatty acids in the microalgae *Spirulina* can be the cause of the distinguishing characteristics of *Spirulina* on the reduce of serum lipid levels. The GC-MS analysis of the *S. platensis* methanolic extract shows fatty acids that have also been reported to have some anti-cholesteremic activity, especially eicosenoic acid and high-oleic safflower oil (sunflower oil is mainly triglycerides, typically derived from the fatty acids linoleic acid "poly-unsaturated omega-6" and "oleic acid" "mono-unsaturated omega-9" with differing concentrations). Hexadecanoic acid (palmitic acid) ethyl ester, exists naturally in butter, cheese, milk, and meat, as well as cocoa butter, soybean oil, and sunflower oil; also

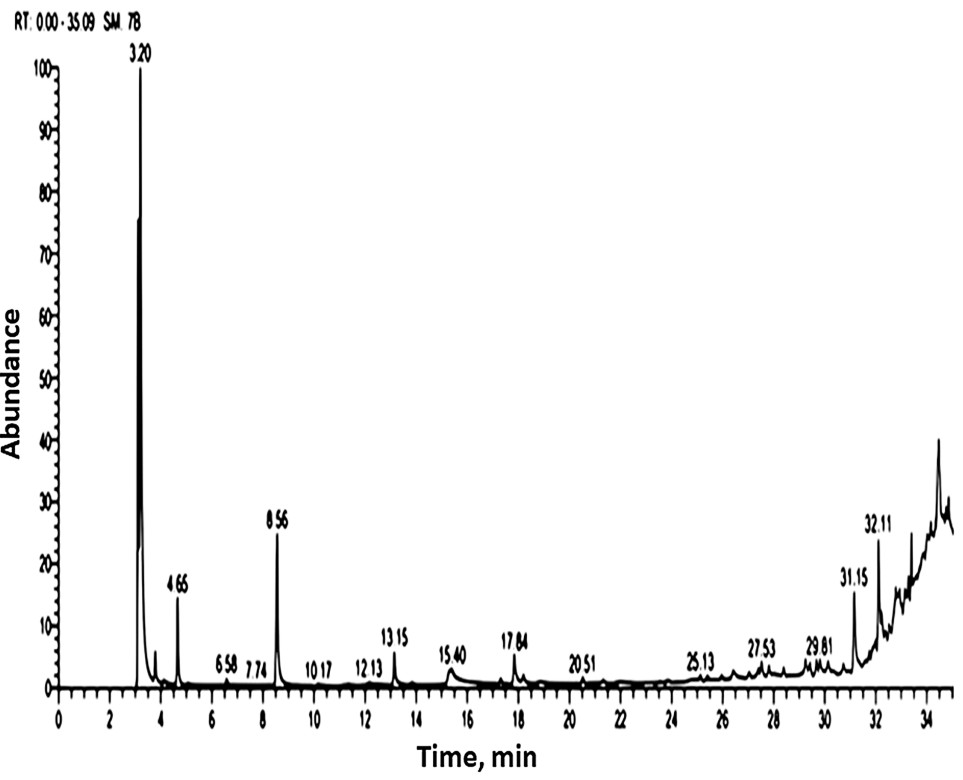

**Figure 2 Gas chromatography and mass spectrometry spectrum of methanolic extract of *S. platensis* (TIC: total ion chromatogram).**

**Table 5 Relative weight gain of the control and experimental animals of male Swiss Albino (SA) mice fed a high-cholesterol diet and different *S. platensis* formulations for 5 weeks.**

| Animal group/treatments | | Initial weight (g) | Final weight (g) | Body weight gain | |
|---|---|---|---|---|---|
| | | | | g | % |
| I | Control | $30 \pm 0.42^{a*}$ | $34.00 \pm 0.38^{a}$ | $4.00 \pm 0.23^{a}$ | 13.33 |
| II | *S. platensis* (15 mg/mL) | $29 \pm 0.21^{a}$ | $24.68 \pm 0.12^{b}$ | $-4.32 \pm 0.24^{b}$ | −14.90 |
| III | Cholesterol gm/Kg | $30 \pm 0.12^{a}$ | $32.00 \pm 0.50^{a}$ | $2.00 \pm 0.10^{a}$ | 6.66 |
| IV | Cholesterol + *S. platensis* (15 mg/mL) | $31 \pm 0.24^{a}$ | $23.66 \pm 0.42^{b}$ | $-7.34 \pm 0.45^{c}$ | −23.68 |
| V | Cholesterol + atorvastatin (10 mg/Kg) | $30 \pm 0.39^{a}$ | $26.53 \pm 0.21^{b}$ | $-3.47 \pm 0.50^{b}$ | −11.57 |
| VI | Cholesterol + *S. platensis* (7.5 mg/mL) + atorvastatin (5 mg/Kg) | $28 \pm 0.18^{a}$ | $23.49 \pm 0.34^{c}$ | $-4.51 \pm 0.20^{b}$ | −16.11 |

Notes:
* Data are expressed as mean ± standard deviation.
Values in the same column with different superscript letters are significantly different ($p < 0.05$).

sodium palmitate is authorized as a natural additive in organic products (US Soil Association standard 50.5.3).

## Evaluation of body weight and lipid profile

The body weights of mice that received *S. platensis* water extracts are listed in Table 5. It showed that the mice fed on high fats diet increased in weight after 5 weeks of treatment

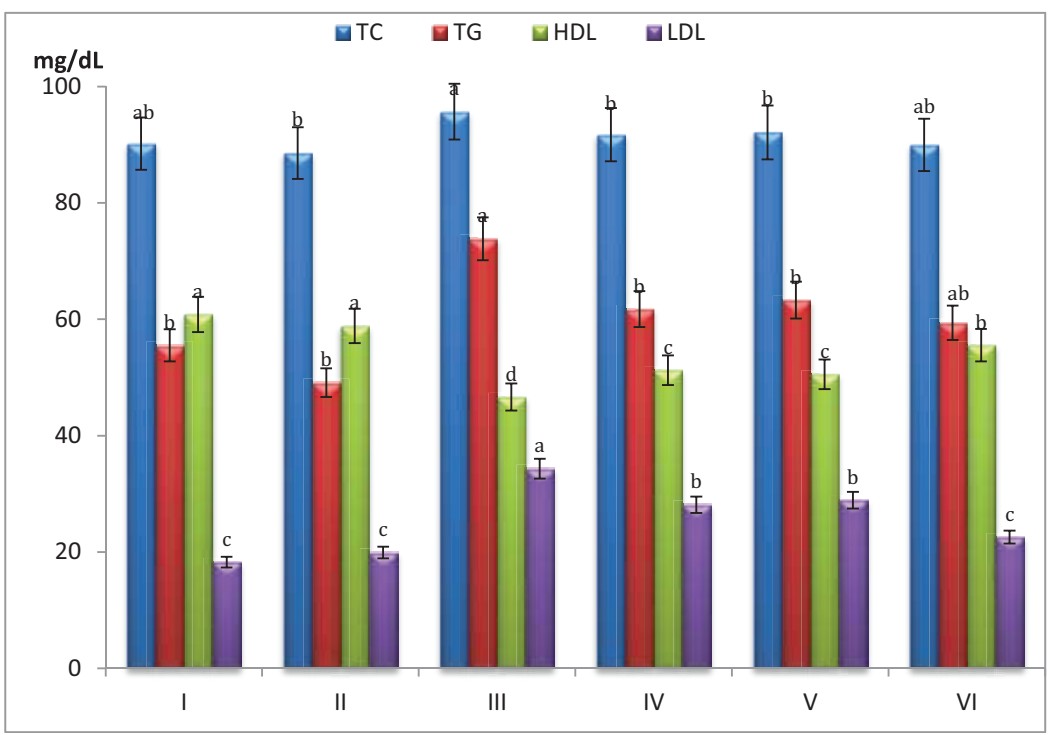

**Figure 3 Effect of different *S. platensis* formulations on plasma cholesterol, triglycerides, cholesterol-high density lipoprotein (HDL) and cholesterol-low density lipoprotein (LDL) in mice after 5 weeks.**

and the mice injected with (15 mg/mL) oral gavage with *S. platensis* only decreased in weight from 29.0 g at the beginning of experiment to 24.7 g by 14.9%.

The mice fed on high fat diet with the same concentration of *S. platensis* decreased from 31.0 g at the beginning to 23.7 g at the end of the experiment by 23.7%, while the mice fed on high fat diet with *S. platensis* in concentration of 7.5 mg/mL + atorvastatin (5 mg/Kg body weight) decreased from 28.0 g at the beginning to 23.5 g at the end of the experiment by 16.1%.

Figure 3 showed that mice fed on high fat diets showed significant increase ($p < 0.05$) in total cholesterol (TC), triglycerides (TG) and LDL and significant decrease ($p < 0.05$) in HDL compared to those fed on normal diet. After 5 weeks of mice treatment with *S. platensis* only, serum TC, TG and LDL levels were reduced significantly ($p < 0.05$), while HDL showed significant increase ($p < 0.05$) as compared to before treatment.

Hypercholesterolemic mice treated with *S. platensis* (7.5 mg/mL) + atorvastatin (5 mg/Kg) showed significant decrease ($p < 0.05$) in serum TC, TG and LDL levels (89.9, 59.4 and 22.5 mg/dL, respectively) and significant increase in the HDL cholesterol level 55.5 mg/dL ($p < 0.05$) compare with control. Also it is clear that the HDL contents were greater in all treatments of *S. platensis* groups especially *S. platensis* only group since it reached 58.8 mg/dL.

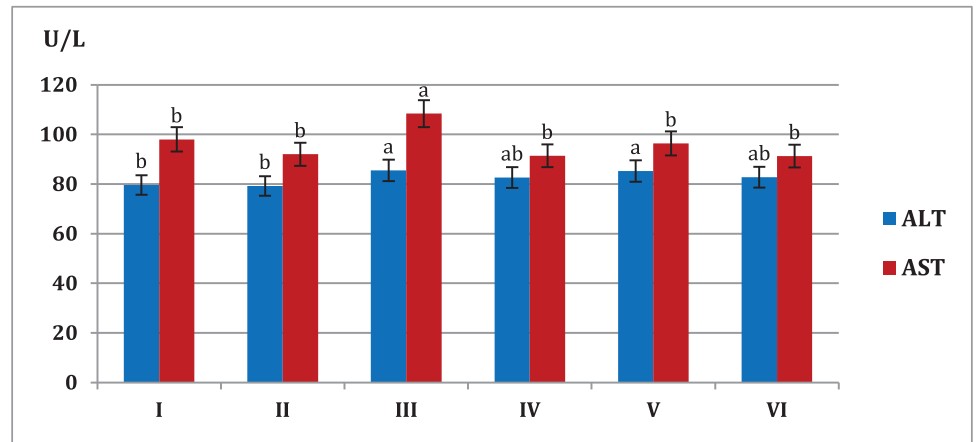

**Figure 4 Effect of different *S. platensis* formulations on liver enzymes; aspartate aminotransferase (AST) and alanine aminotransferase (ALT) in mice after 5 weeks.**

## Evaluation of liver and kidney functions

Figure 4 showed that mice fed on high fat diets showed significant increase ($p < 0.05$) in AST and ALT liver enzymes (108.4, 85.5 U/L, respectively) compared to animals fed on normal diet and *S. platensis* diet only (92.0, 79.2 U/L, respectively). On the other hand liver enzymes significant decreased ($p < 0.05$) in hypercholesterolemic mice which treated with *S. platensis* only or in combination with atorvastatin.

Toxic renal effects are normally manifested with increases in the level of serum urea. The recorded values of renal parameters (urea and creatinine) indicate significant reduction, for the all trials of *S. platensis* with mice (Fig. 5). Consequently, group III showed an increase in the level of renal values, while groups IV, V and VI showed reduction in the level of renal values. On the other hand, groups I and II were considered normal cases.

## Histological study

Normal saline (negative control, group I): liver showed normal hepatic parenchyma with preserved lobules, cords, sinusoids, bile canaliculi, portal area and stromal structures. The kupffer cells were prominent. Heart sections revealed normal cardiomyocytes with a few hyaline degeneration in some of them (Fig. 6). *S. platensis* only (group II): liver showed normal hepatic parenchyma with preserved lobules, cords, sinusoids, bile canaliculi, portal area and stromal structures with prominent kupffur cells. Heart sections revealed mild interstitial edema and myocardial degeneration (Fig. 7). The mice fed on high fat diets (positive control, group III): liver sections showed wide spread centrolobular and periportal micro and macrosteatosis beside portal round cells aggregation. Heart sections revealed mild congestion of the coronary blood vessels and intermuscular capillaries. Moderate number of cardiomyocytes showed macrosteatosis. Interstitial edema was also seen (Fig. 8). The mice fed on high fat diets with *S. platensis* (group IV): liver sections showed wide spread centrolobular and periportal micro and macro steatosis.

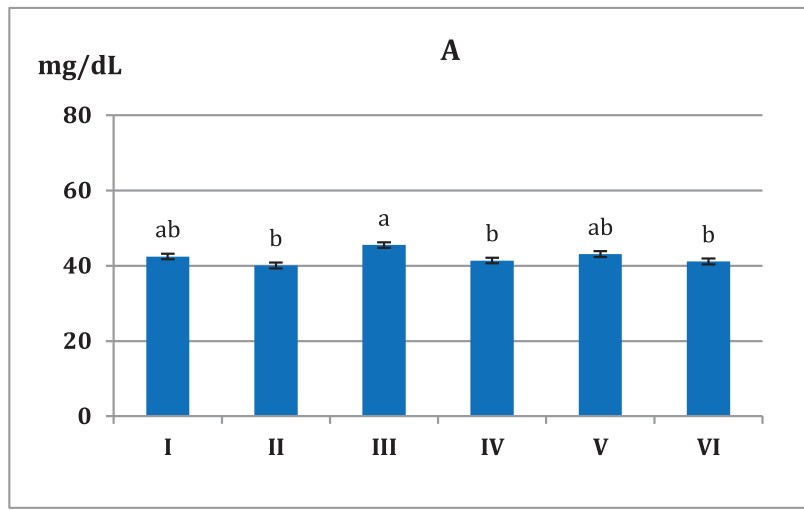

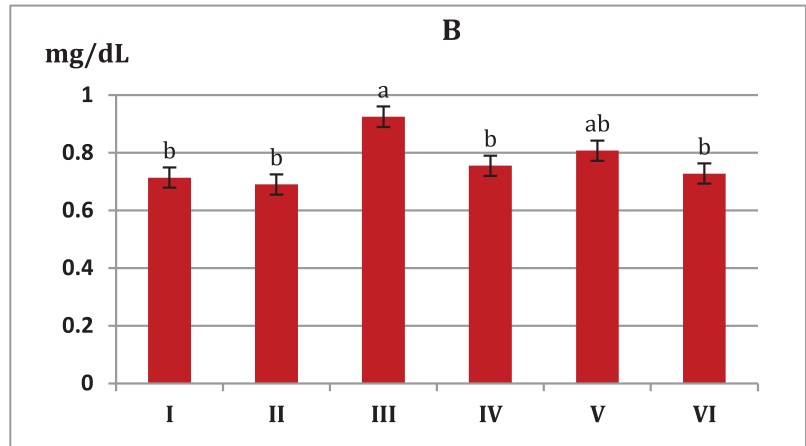

**Figure 5 Effect of different *S. platensis* formulations on kidney functions; urea (A) and creatinine (B) in mice after 5 weeks.**

Mild portal round cells aggregations and prominent kupffer cells were observed. Heart: showed mild congestion of the coronary blood vessels and intermuscular capillaries. Mild to moderate number of cardiomyocytes showed macrosteatosis. Interstitial edema was also seen (Fig. 9). The mice fed on high fat diets with atrovastin 10 mg/Kg instead of *S. platensis* (group V): liver sections showed congestion of hepatic blood vessels with perivascular leukocytic infiltration mainly neutrophils in addition to extravasated erythrocytes. Examined sections from the heart showed sever congestion of coronary blood vessels, interstitial edema beside steatosis and hyaline degeneration of some cardiomyocytes (Fig. 10). The mice fed on high fat diets with atrovastin 5 mg/Kg and *S. platensis* 7.5 mg/Kg (group VI): liver sections revealed apparently normal hepatic parenchyma with preserved lobules, cords and sinusoids beside presence some dispersed apoptotic and de-generated cells. Heart sections revealed normal cardiomyocytes with mild interstitial edema and hyaline degeneration in some cells (Fig. 11).

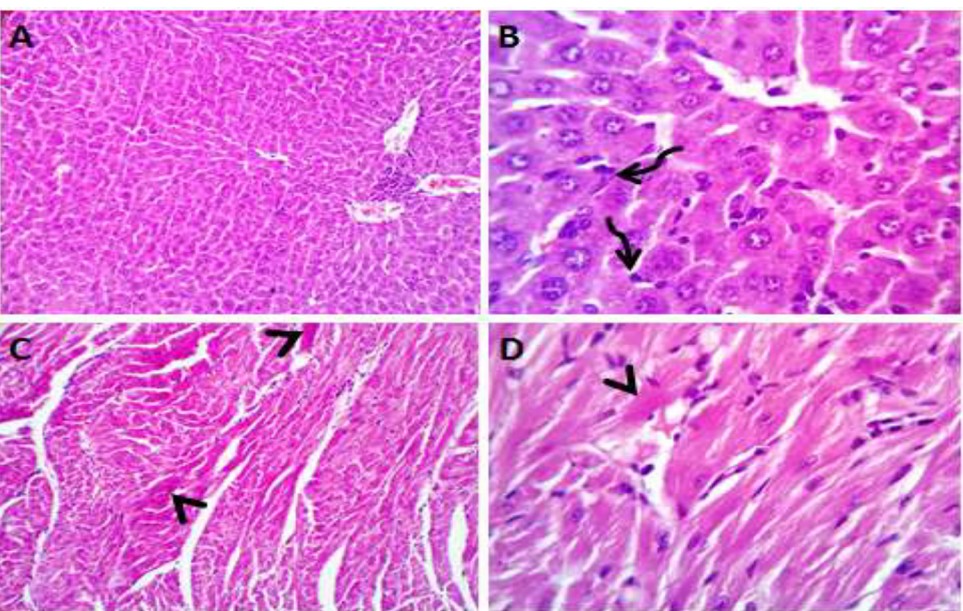

**Figure 6** Photomicrograph of liver sections (A and B) showing normal hepatic parenchyma with preserved hepatic cords, sinusoids and prominent kupffur cells (curved arrows); heart sections (C and D) showing normal cardiomyocytes with hyaline degeneration in some cells.

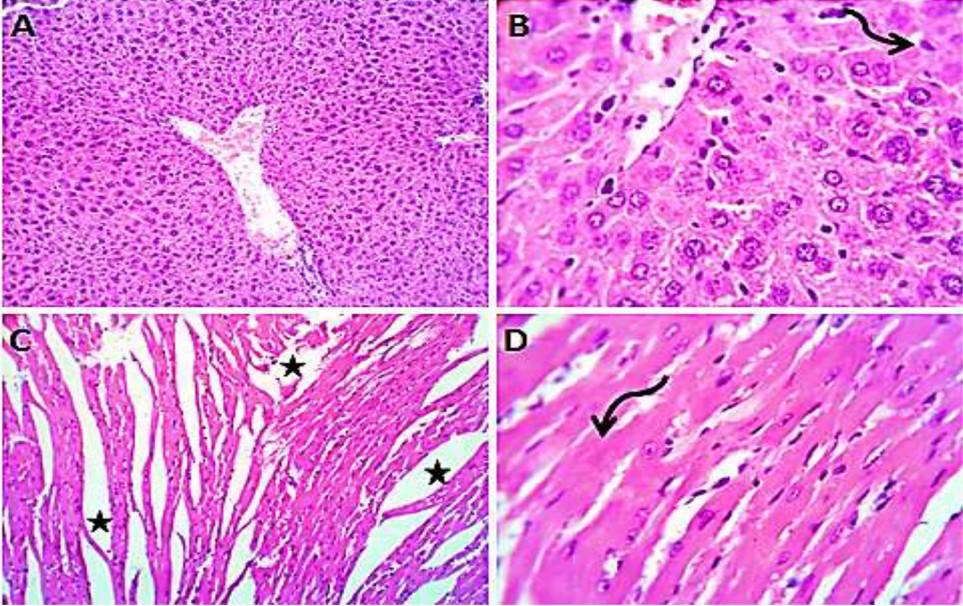

**Figure 7** Photomicrograph of liver sections (A and B) showing normal hepatic cords, sinusoids and prominent kupfffur cells; heart sections (C and D) showing mild interstitial edema (stars) and myocardial degeneration (curved arrow). H&EX 100 (A and C), 400 (B and D).

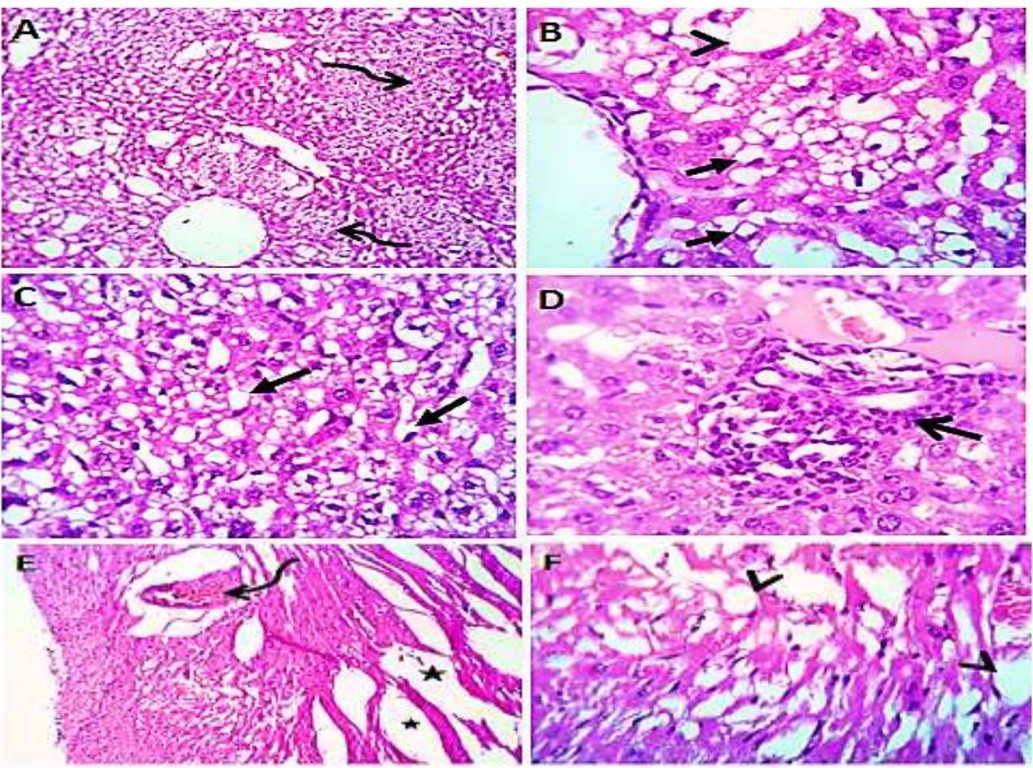

**Figure 8** Photomicrograph of liver (A–D) showing widespread centrolobular (curved arrow) micro (closed arrows) and macro steatosis (arrow head) beside portal round cells aggregation (open arrow); heart sections (E and F) show mild congestion of the corona.

## DISCUSSION

Increased pigment concentration (chlorophyll a content and total carotenoid) was closely related to cell dry weight of *S. platensis*. Chlorophyll is one of the precious bioactive compounds that can be obtained from microalgae biomass. Chlorophyll is used as antioxidant, natural food coloring agent and has anti-mutagenic properties (*Priyadarshani & Rath, 2012*). Carotenoids are used in feed and food as flavorings and colorants and in nutritional supplements as a source of pro-vitamin A (*Kim, 2015*). There is confirmation that these pigments may protect humans from dangerous disorders related with oxidative and inflammatory stress involving skin degeneration, CVD, aging, certain types of cancer, and age-related diseases of the eye (*Rao & Rao, 2007*).

*Zeng et al. (2001)* reported that the characteristics of light had effect on the growth of *S. platensis* cells. *Habib et al. (2008)* introduced average irradiance and specific light energy utilization rate to describe the relationship between cell growth and light utilization. *Olaizola & Duerr (1990)* found that *Spirulina* can respond to changes in the spectral distribution of growth irradiance by changing its chlorophyl a and carotenoid content. Chlorophyll absorbs red and blue light best, so algae whose cells contain high concentration of chlorophyl if are cultivated in red and blue flasks, grow well and tend to produce a large amount of biomass; compared to those under green flask conditions.

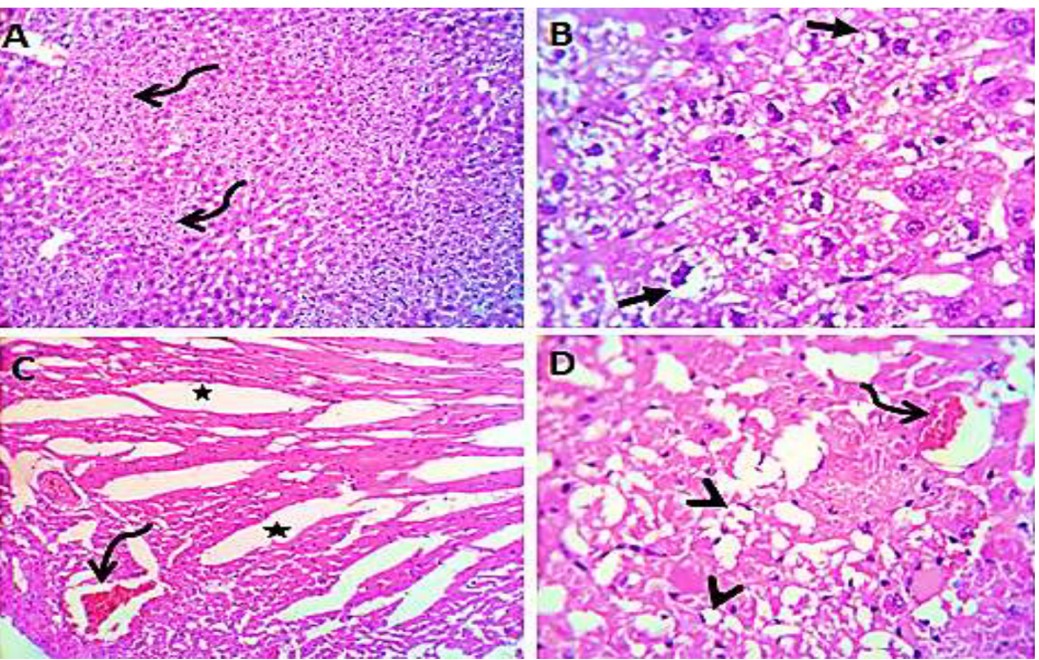

**Figure 9** **Photomicrograph of liver showing widespread centrolobular (curved arrow) micro steatosis (closed arrow).** Heart sections (C and D) showing mild congestion of the coronary blood vessels (curved arrows), interstitial edema (stars) and micro steatosis (arrow heads) in a moderate number of cardio-myocytes. H&EX 100 (A and C), 400 (B and D).

In this case, the photoreceptor called cryptochrome (a sensor of blue irradiance and blue/green ratio) might be responsible for inhibition (*Sellaro et al., 2010*).

*Olaizola & Duerr (1990)* concluded that the growth rate of *S. platensis* increased rapidly under white and red lights, while under blue light the growth rate obtained lower; Also, *Vo et al. (2017)* reported that the red and white light conditions induced the growth and biosynthesis organic constituents for instance carotenoid.

From the above results we could consider *S. platensis* in oceans and in open area to be very useful mass as a dry weight, so the effect of light so important; not cancelation the light or making a barrier between light and the microorganisms. *Colla, Muccillo-Baisch & Costa (2008)* affirmed that the presence of antioxidant compounds such as phenolic acids, a-tocopherol and b-carotene presents in *Spirulina* extracts were responsible for the antioxidant activity and can be the cause of the properties of *Spirulina* on the decrease of serum lipid levels. *Nagaoka et al. (2005)* found that both, *S. platensis* concentrate (SPC) and C-phycocyanin (a proteic pigment extracted of *Spirulina*) caused hypocholesterolemic effect in rats; also, proposed that the hypocholesterolemic mechanism of SPC may involve the inhibition of both jejunal cholesterol absorption and ileal bile acid reabsorption, furthermore C-phycocyanin might be the active ingredient in *Spirulina* responsible for the hypolipidemic activity than SPC. Oleic acid linoleic and palmitic acids were found in methanolic extracts of *Spirulina*. Oleic acid is associated with several health benefits, which mainly include anti-cancer activity and prevention of CVD, platelet aggregation and hypertension (*De Souza et al., 2015*). *Singh & Nimbkar (2016)*

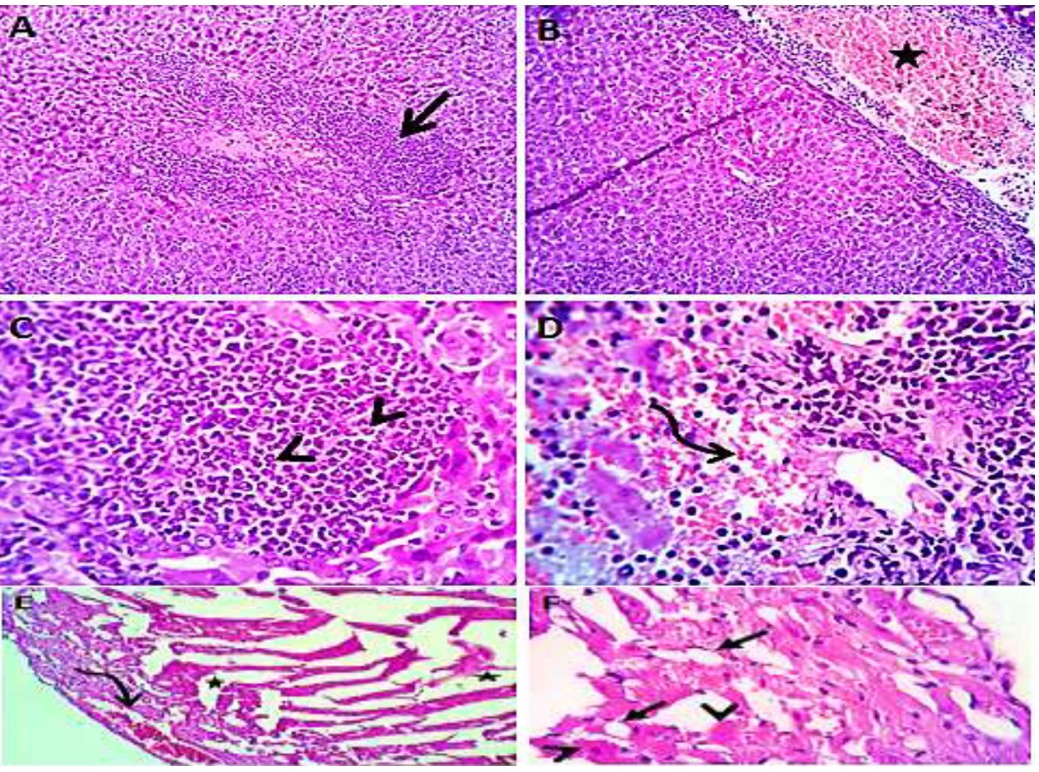

**Figure 10** Photomicrograph of liver (A–D) showing congestion of hepatic blood vessels (star) with perivascular leukocytic infiltration (open arrow) mainly neutrophils (arrow head) in addition to extravasated erythrocytes (curved arrow). Sections from the heart (E and F) showing sever congestion of coronary blood vessels (curved arrow), interstitial edema (stars) beside steatosis (closed arrows) and hyaline degeneration (arrow heads) of some cardiomyocytes. H&EX 100 (A and B), 400 (C, D, E and F).

testified that an eight gram daily dose of safflower oil for 16 weeks can enhance health measures such as glycemia, inflammation, and blood lipids in obese, postmenopausal women who have diabetes type 2. Our results agree with those reported by *Gheda, Khalil & Gheida (2013)* who found that the higher percentage of linoleic and palmitic acids were found in methanolic extracts of *Spirulina* compared to crude *Spirulina*. These results open our minds for the natural medications to diminish weight and decrease cholesterol level within few weeks. Similar finding was obtained by *Zhao et al. (2019)* who stated that *S. platensis* protein (SPP) and protein hydrolysate (SPPH) possessed anti-obesity effect in mice and total cholesterol reduction activities by modulating the expressions of various key genes related with lipid metabolism in brain and liver, for instance Acadm, Retn, Fabp4, Ppard, Slc27a1, etc. The main metabolic pathways for reducing cholesterol are via conversion to bile acids or preventing the cholesterol synthesis by inhibiting the HMG CoA reductase enzyme; *Ama Moor et al. (2017)* found that *S. platensis* may reveal hypolipidemic activity through the activation of lecithine cholesterol acyl transferase LCAT, also confirmed that the administration of *S. platensis* inhibits the plasmatic activity of HMG COA reductase significantly in rats. *Belay (2002)* also stated that administration of *Arthrospira maxima* associated with simvastatin prevents the acute fatty liver

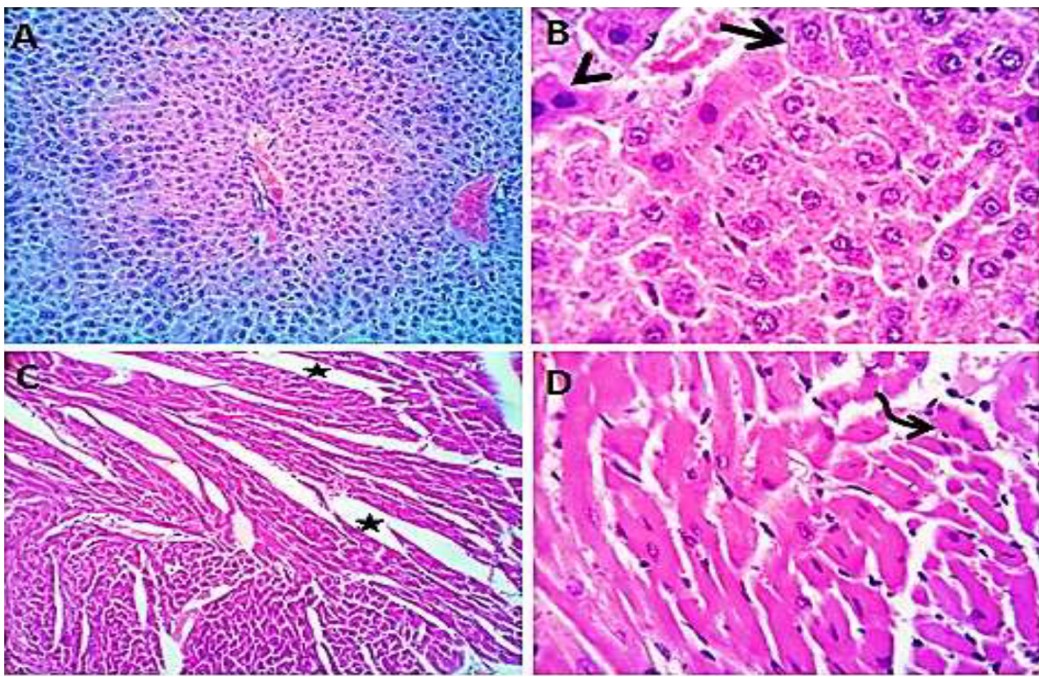

**Figure 11 Photomicrograph of liver (A and B) showing apparently normal hepatic parenchyma with preserved lobules, cords and sinusoids beside presence some apoptotic (arrow head) and degenerated (open arrow) cells; heart sections (C and D) showing normal cardiomyocytes with mild interstitial edema (stars) and hyaline degeneration in some cells (curved arrow).** H&EX 100 (A and C), 400 (B and D).               

primary approach for controlling de novo cholesterol synthesis. *Torres-Duran, Ferreira-Hermosillo & Juarez-Oropeza (2007)* concluded that *S. maxima* had hypolipemic effects, especially on the TG and the LDL-C concentrations, but indirectly on TC and HDL-C values, and positive effects on reducing systolic and diastolic blood pressure by enhancing nitrous oxide release. *Colla, Muccillo-Baisch & Costa (2008)* found that the serum levels of HDL were higher in the groups fed with *Spirulina*, and emphasized the importance of *Spirulina* biomass reducing the serum levels of total cholesterol and increasing the serum levels of HDL-cholesterol.

## CONCLUSIONS

Artificial light provides better regulation of intensity, duration, and light spectra and light characteristics have a profound influence on microalgae metabolism and development. The present study provides a new insight for production of blue-green algae *S. platensis* biomass in different colored flasks under artificial illumination, also the anti-hypercholesterolemic potential of aqueous extract of *S. platensis* in vitro and in the diet induced hypercholesterolemic Swiss albino mice. *S. platensis* developed in colorless flasks under continuous illumination using cool white fluorescent with the intensity of photosynthetically light was 2,500 lux (35 µmol m$^{-2}$ s$^{-1}$) demonstrated the highest chlorophyl a and carotenoid contents based on dry weight, furthermore administration of hot aqueous extract of *S. platensis* which obtained from the same kind of flasks resulted

in vitro and in vivo hypolipidemic effects. The highest cholesterol reduction percentage in vitro (76.01%) was observed using the hot aqueous extract of *S. platensis* at a concentration of 15 mg/mL after 60 min of incubation; also clearly showed that extract produced from *S. platensis* reduced serum TC, TG and LDL; furthermore, it increased serum HDL levels and possessed anti-obesity effect in animal model.

In conclusion, the hot aqueous extract of *S. platensis* developed in colorless flasks has great potential as possible therapy for reducing cholesterol levels. This product exerted a significant hypocholesterolemic effect on Swiss albino mice fed with a high cholesterol diet. *S. platensis* consumption as a dietary supplement might be useful in reducing total cholesterol and TG levels in the serum and liver for humans. Hot aqueous extract of *S. platensis* appears to be safe for its potential use in hypercholesterolemia control.

## Scope for future work

Microalgae are among the fastest growing photosynthetic organisms giving a higher yield than other crops they have unique properties of $CO_2$ fixation and release of oxygen. The research findings and trials of this paper can be a fillip for further research on production of low cost *Spirulina* with superior quality via optimizing physical and chemical conditions of production medium; moreover further clinical studies are needed for confirming some pharmaceutical properties such as anti-diabetic and cancer activities.

### Funding
The authors received no funding for this work.

### Competing Interests
The authors declare that they have no competing interests.

### Author Contributions
- Mahmoud A. Al-Saman conceived and designed the experiments, analyzed the data, prepared figures and/or tables, authored or reviewed drafts of the paper, and approved the final draft.
- Nada M. Doleib conceived and designed the experiments, authored or reviewed drafts of the paper, and approved the final draft.
- Mohamed R. Ibrahim conceived and designed the experiments, performed the experiments, prepared figures and/or tables, and approved the final draft.
- Mohamed Y. Nasr conceived and designed the experiments, prepared figures and/or tables, authored or reviewed drafts of the paper, and approved the final draft.
- Ahmed A. Tayel conceived and designed the experiments, analyzed the data, authored or reviewed drafts of the paper, and approved the final draft.
- Ragaa A. Hamouda conceived and designed the experiments, analyzed the data, prepared figures and/or tables, authored or reviewed drafts of the paper, and approved the final draft.
## Animal Ethics

The following information was supplied relating to ethical approvals (i.e., approving body and any reference numbers):

The Ethics Committee of the Genetic Engineering and Biotechnology Research Institute, University of Sadat City, Sadat City, Egypt approved this study (approval number: gebri USC-009-1-19).

## Field Study Permissions

The following information was supplied relating to field study approvals (i.e., approving body and any reference numbers):

Approval by Genetic Engineering and Biotechnology Research Institute, University of Sadat City, Sadat City, Egypt (gebriUSC-009-1-19).

## Data Availability

The raw data are available in the Supplemental Files.

## Supplemental Information

Supplemental information for this article can be found online at http://dx.doi.org/10.7717/peerj.10366#supplemental-information.

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
