# Peer review of "In vitro and in vivo hypolipidemic properties of the aqueous extract of Spirulina platensis, cultivated in colored flasks under artificial illumination"

_PeerJ, doi:10.7717/peerj.10366_

## Round 0.1 · original submission · Minor Revisions

Dear Dr. Hamouda,

Three reviewers provided a list of minor but extensive requests that will make your manuscript stronger. Please revise your MS in accordance with the reviewers' suggestions, paying special attention to providing a more detailed method section and a deeper discussion.

Looking forward to a revised version of your work.

Reviewer 1 ·

Basic reporting

The objective of the work may clearly be stated at the end of the introduction.
The detailed methodology of the work may be presented in a Flowchart form
The literature review may include previous works on similar topic .The following may be useful .
“Spirulina – From growth to nutritional product: A review,” Trends Food Sci. Technol., vol. 69, 2017, doi: 10.1016/j.tifs.2017.09.010.
Comparative study on the growth performance of Spirulina platensis on modifying culture media,” Energy Reports, vol. 5, 2019, doi: 10.1016/j.egyr.2019.02.009.

Experimental design

Line 122 justify the absorbance measurements at three different wavelengths 663, 645 and 480 nm.and its physical significance.
Indicate unit g /L instead of mg/mL
Provide the equation for the growth rate of spirulina.
Page 35: Indicate results discussed either in tabulated form on graphically to understand effect of each process variable. A comparison of the data with literature should also be done.Comment on mechanism of Spirulina cultivation with time.
What is the unit of SP growth rate, explain and check units.
Author should provide quantitative data for growth rate and biomass content at different cultivation conditions.
Compare the results obtained with published literature and discuss.
There is more variation in SP growth rate (Table 3) with colorless than color . Explain
Page 97: What information can be derived from GC-MS. This has to be discussed. What is typical chemical composition of algae and what could be the possible application other than the work reported in this study .

Validity of the findings

What are the contribution of work towards global community and audience future deployment of spirulina cultivation
The conclusions need to be re-written in point by point form with key numbers obtained from the study. Conclusions seems to be weak.It should be extended .
No reports are available on cultivation of blue-green algae Spirulina platensis in
399 Erlenmeyer flasks of different color under artificial illumination.Justify.
Similarly future scope should be focused.

Additional comments

The paper titled “In vitro and in vivo hypolipidemic properties of the aqueous extract of Spirulina platensis, cultivated in colored flasks under artificial illumination” is very interesting and within the scope of the journal.The above suggestions are provided to improve the quality of the manuscript.

Reviewer 2 ·

Basic reporting

Lack of literature findings

Experimental design

Investigation is to be improved by varying other parameters inaddition to the coloured flask variation

Validity of the findings

Conclusion is not well defined and key findings are not represented. Sprirulina is one of the richest protein sources in microalgae. How much protein content in dry weight it would contain.

Reviewer 3 ·

Basic reporting

The work of Al-Saman et al., entitled “In vitro and in vivo hypolipidemic properties of the aqueous extract of Spirulina platensis, cultivated in colored flasks under artificial illumination” examined the effect of hot water extract of S. platensis grown in colored glass bottles on cholesterol levels reduction in vitro and in vivo. In general the manuscript is well prepared and outlined. However, some points must be revised for a better understanding of the study. The considerations follow below:

Abstract
1. Add the aim of the study to the summary.
2. There is a lot of information missing from the methods section. For example, what was the biological model used? What are is the characteristics of the model (rats, mice..)? What route of administration was used? What dose administered and for how long?
3. It is missing a conclusion in the summary.
Introduction
1. In the first paragraph of the introduction (line 56-57), please remove the following sentence "Cholesterol can be both good and bad."
2. The references used to support spirulina are very old, especially those related to composition, such as Bujard et al., 1970; Habib et al., 2008. Are there no more current references?
3. The purpose of the work should be made clearer.

Experimental design

Materials and methods
1. What was the final concentration used for the in vitro anti-cholesterol assay test?
2. Why did they use two-week-old young mice, not adult mice?
3. The concentration of 15 mg / mL was based on what?
4. Please reference the protocol methodology in vivo.
5. Why, when combined, atorvastatin and spirulina had their dose reduced?
6. How is the glucose level of the animals?

Validity of the findings

Results and discussion
1. Have any in vitro cytotoxicity before been performed for further in vivo studies?
2. In the "In vitro cholesterol-lowering activity of Spirulina platensis cultivated in colored flasks" section, in the last paragraph. Please explain how the antioxidant capacity decreases serum lipid levels.
3. The discussion about the influence of the extract on the lipid profile must be deepened.
4. In the description of body weight results, please add the value of p.
5. In order to better understand the cardioprotective role of the extract, if possible, it would be extremely important to investigate the activity of paraoxonase-1, since it plays an important antioxidant role associated with HDL. Otherwise, the addition of a possible mechanism of action could be added in the discussion.

Additional comments

No comment

Annotated reviews are not available for download in order to protect the identity of reviewers who chose to remain anonymous.

Reviewer 4 ·

Basic reporting

• Does Spirulina platensis grow in the country of the authors ? is it harvested in some farms ?
• The english should be improved
• Why are the references Ishikawa and Jeffrey not in bold ?
• We did not understand well the use of letter a, b, c,d representing the statistical difference. It should be well explain.

Experimental design

• Why the authors did the culture in flask ? we think that it is important to tell us what are the challenge with the culture of Spirulina platensis in their country. It is harvested elsewhere.
• They did not test the presence of the major pigment of Spirulina platensis, phycocyanin
• Line 129 : please do not begin the sentence with a number. See how it is done in line 132
• Line 142 : powder or power
• Line 156 : what is EI mass spectra ?
• Line 171 : and each group containing 8 mice
• Line 179 : ‘’of 1 g cholesterol for each 1 kg diet plus from S. platensis’’ . It seems to be a non sense
• Line 180 : why not the same concentration of spirulina and atorvastatine like in other group. The concentration of Spirulina and atorvastatine are divides by 2
• Line 187 : Where the test tube dry ? or with any anticoagulant ?

Validity of the findings

• Lines 365 to 367 : what does that means ? toxic renal effects are only revealed by the dosage of urea ?
• Line 392 : is it spirulina 7.5 mg/ml ? or 75 mg/kg ?
• Does the presence of proeminent Kupffer celles, edema in the liver means inflammation ?
• In the heart, what the congestion of coronary vessels suggest ?

Additional comments

• We propose to harmonise the spelling of chlorophyll a : See line 38 and line 42
• Line 56 : cholesterol plays an essentiel role. Not the essential role
• Is cholesterol only essential for heart ? We suggest to put 2 or 3 roles of cholesterol
• Line 60 : major cause not chief cause
• The english will have to be improved . Ex : lines 66, 67, 71,72,73,74. The traduction is not well done
• Line 76 : this allowed… not led
• Line 82 : teh reference are too old. There are new references in Africa
• Line 90 : there a in vivo and in vitro studies done in Africa on the hypolipidemic effects of Spirulina platensis

---

## Round 0.2 · accepted · Accept

Congratulations on the improvement on your MS. Please, during the final checks, please consider including the suggestion of reviewer 4. Kind regards,

Reviewer 1 ·

Basic reporting

no comment'

Experimental design

no comment'

Validity of the findings

no comment'

Reviewer 4 ·

Basic reporting

Nothing to say

Experimental design

Nothing to say

Validity of the findings

Line 368 : what does that means ? toxic renal effects are only revealed by the dosage of urea ?
Here is the response of the authors : Mice were not treated with urea; but the toxic renal effects associated with increased serum urea and creatinine levels

Please add creatinine in the sentence . We want to let them know that evaluation kidney function is explored by the two, urea and creatinine, and mainly by the glomerular filtration rate as recommended by kidney societies

Additional comments

Nothing to say